# Ciprofloxacin Concentrations in Food Could Select for Quinolone Resistance in *Klebsiella pneumoniae*: An In Vivo Study in *Galleria mellonella*

**DOI:** 10.3390/antibiotics13111097

**Published:** 2024-11-18

**Authors:** Nele Panis, Zina Gestels, Dorien Van Den Bossche, Irith De Baetselier, Said Abdellati, Thibaut Vanbaelen, Tessa de Block, Sheeba Santhini Manoharan-Basil, Chris Kenyon

**Affiliations:** 1STI Unit, Department of Clinical Sciences, Institute of Tropical Medicine, 2000 Antwerp, Belgium; nele.panis8@gmail.com (N.P.); zgestels@itg.be (Z.G.); tvanbaelen@itg.be (T.V.); sbasil@itg.be (S.S.M.-B.); 2Clinical and Reference Laboratory, Department of Clinical Sciences, Institute of Tropical Medicine, 2000 Antwerp, Belgium; dvandenbossche@itg.be (D.V.D.B.); idebaetselier@itg.be (I.D.B.); sabdellati@itg.be (S.A.); tdeblock@itg.be (T.d.B.); 3Division of Infectious Diseases and HIV Medicine, University of Cape Town, Cape Town 7700, South Africa

**Keywords:** *K. pneumoniae*, ciprofloxacin, AMR, MSC (minimal selective concentration), *G. mellonella*, ADI (acceptable daily intake)

## Abstract

**Background**: The use of antimicrobials to treat food animals leaves antimicrobial residues in foodstuffs. The World Health Organization (WHO) defines the acceptable daily intakes (ADIs) of these residues as the dose of these antimicrobials that is safe for an average human to consume on a daily basis. We hypothesized that the lowest dose of ciprofloxacin classified as safe by the WHO could select for ciprofloxacin-resistant strains of *Klebsiella pneumoniae* in a *Galleria mellonella* model. **Objectives**: We aimed to evaluate if the consumption of peri-ADI doses of ciprofloxacin could select for ciprofloxacin-resistant (Ser464Phe, GyrB, ciprofloxacin MIC of 4 µg/mL) compared to -susceptible (isogenic, ciprofloxacin MIC of 0.047 µg/mL) strains of *K. pneumoniae* in a *Galleria mellonella* model. **Results**: A significant increase was seen in the proportion of resistance for the 1× ADI and 1/10th ADI concentrations on day 2 compared to the positive control. **Methods**: A model of *K. pneumoniae* infection in *G. mellonella* larvae was used for the experiment. The larvae were inoculated with *K. pneumoniae* followed by 10× ADI, 1× ADI, 1/10th ADI, 1/100th ADI, and 1/1000th ADI doses of ciprofloxacin. The isolation of *K. pneumoniae* colonies was then performed on selective agar plates with and without ciprofloxacin (1 µg/mL). The proportion of colonies with ciprofloxacin resistance was then calculated for each group at 24 and 48 h. **Conclusions**: We found that, at 48 h, there was an enrichment of *K. pneumoniae* colonies with ciprofloxacin resistance in the larvae receiving 1× ADI and 1/10th ADI concentrations of ciprofloxacin. These results suggest that the ciprofloxacin MSC_select_ for *K. pneumoniae* in this model is 1/10th of the acceptable daily concentration (ADI) dose of ciprofloxacin, which is equivalent to 0.239 ng/µL.

## 1. Introduction

In 2011, Gullberg et al. found that ciprofloxacin concentrations were 230-fold lower than the minimum inhibitory concentration (MIC), selected for ciprofloxacin-resistant strains of *Escherichia coli* [1]. He termed the lowest concentration of an antimicrobial that could select for antimicrobial resistance (AMR) to be the minimum selection concentration (MSC) [1]. Two types of MSC have been defined [2]. The MSC_denovo_ is defined as the minimum concentration of an antimicrobial at which one can induce de novo resistance [1,2,3]. The MSC_select_ is the lowest antimicrobial concentration that selects for a resistant compared to a susceptible strain [1,2,3,4]. All Gullberg’s MSC experiments were performed in vitro [1,5].

More recently, studies have established the *Galleria mellonella* infection model as a useful way to ascertain the in vivo MSC_denovos_. One of these studies has established that the in vivo ciprofloxacin MSC_denovo_ for *Klebsiella pneumoniae* is 1/10th of the dose that the European Medicines Association (EMA) defines as an acceptable daily intake (ADI) for humans [6]. Likewise, another study found that the in vivo *Streptococcus pneumoniae* MSC_denovo_ was 1/10th of the concentration defined as safe by the EMA [7]. Whilst lower than ADIs, these in vivo MSCs were considerably higher than the equivalent in vitro MSCs [1]. MSC_selects_ tend to be considerably lower than MSC_denovos_. Gullberg, for example, found the MSC_select_ for ciprofloxacin to be approximately 23-fold lower than the MSC_denovo_ [5].

Nonetheless, the World Health Organization (WHO) and other authorities do not include data on the minimum selective concentration in their safety measurements [8]. Therefore, in this paper, we assess the in vivo ciprofloxacin MSC_select_ of *K. pneumoniae*. To the best of our knowledge, no study has yet assessed an MSC_select_ in vivo. We use the standard *G. mellonella* in vivo model to conduct these experiments. The ciprofloxacin concentrations we test are based on the World Health Organization (WHO)/Food and Agriculture Organization (FAO) definition of an ADI. The ADI is defined by the FAO/WHO as “an estimate of the amount of a food additive in food or beverages expressed on a body weight (bw) basis that can be ingested daily over a lifetime without appreciable health risk to the consumer” [9,10,11]. The WHO/FAO use the ADIs and other information such as dietary exposure to the relevant foodstuff to set the maximum residue limits (MRLs) [12]. The WHO and FAO define the MRL as “the maximum concentration of residue resulting from the use of a veterinary drug (expressed in mg/kg or μg/kg on a fresh weight basis) that is recommended by the Codex Alimentarius Commission (CAC) to be legally permitted or recognized as acceptable in or on a food” [11,12,13]. The MRL is determined as the concentration that ensures that the residues in food do not exceed the ADI [13]. The European Medicines Agency (EMA) determined the enrofloxacin ADI as 6.2 µg/kg [14,15,16].

## 2. Results

### 2.1. Validating the 50:50 Ratio of the Susceptible and Mutant Strains in the Mix

The different colony counts were represented as a scatterplot for the two concentrations (Appendix A). Appendix A shows that the wildtype strain cultured onto plates containing ciprofloxacin from the 10^−5^ dilution showed no colonies. An increase in colony count was seen for the mutant strain compared to the 50:50 mix in the ciprofloxacin plates of concentration 10^−5^ (Appendix A). On the contrary, a decrease in colonies was found for the wildtype strain compared to the 50:50 mix. The mutant and wildtype strains were significant compared to the 50:50 mix (Table 1). The median proportion of the 50:50 mix colonies observed on the ciprofloxacin plate for concentrations 10^−5^ was 0.7 (28/40). For the control plate, a generally higher colony count is observed compared to the mutant and wildtype strains (Appendix A). A median proportion of approximately 0.61 (46/76) was observed with medians of 35, 41 and 46, respectively. However, no significant difference was found comparing the wildtype and mutant strain to the 50:50 mix (Table 1).

At the more diluted concentration of 10^−6^, the number of colonies was approximately ten-fold lower, with similar relations between the three groups compared to those observed at the higher concentration. The colony counts of the 50:50 mix lie in between the wildtype and mutant strain for the ciprofloxacin plates (Appendix A). An equal trend was observed when compared to the concentration of 10^−5^. Furthermore, a significant value of 0.0079 was found for the wildtype strain compared to the 50:50 mix (Table 1). However, no significance was observed for the mutant strain compared to the 50:50 mix (Table 1). The median proportion of the 50:50 mix was calculated to be 0.67 (2/3) for the ciprofloxacin plates. A slightly different trend was noticed for the control plates compared to the concentration of 10^−5^. A median of four colonies was observed for both the wildtype and mutant strains and in the 50:50 mix, and seven colonies were observed on the control plates. A proportion of 0.875 (7/8) was observed for the 50:50 mix on the ciprofloxacin plate. However, no significant difference was seen when comparing both strains to the 50:50 mix (Table 1).

### 2.2. Colonization of the Larvae

*K. pneumoniae* colonization of the larvae was required for two days. To test if the colonization for this duration was successful, colonies that appeared on the selective control plates without ciprofloxacin after hemolymph extraction on days 1 and 2 were counted. These colonies were counted for the positive control and the different ADI concentrations (10× ADI, 1× ADI, 1/10th, 1/100th, and 1/1000th ADI dose). Appendix A shows the distribution of the colony counts for the different ciprofloxacin ADI concentrations and the positive control. The 1/10th ADI concentration showed a significant increase compared to the positive control for day 1 (Table 2). On the other hand, a significant decrease was seen for the 1/1000th ADI concentration compared to the positive control (Table 2). Further, the ciprofloxacin concentrations of 1/10th and 1/100th ADI showed an increase in colony counts compared to the positive control for day 2 (Table 2). The colonies appearing for each different ADI concentration and positive control showed that the colonization was successful. No colonies were seen for the negative controls, containing only PBS.

### 2.3. Mortality Rate

The mortality rate was monitored at 24 and 48 h intervals after injecting the larvae with the *Klebsiella pneumoniae* mix and various doses of ciprofloxacin or PBS (Appendix A). A Kaplan–Meier survival graph was made showing the survival percentage for the various experimental groups on day 1 and day 2. Larvae that looked fully melanized (black) and did not move when pricking with a toothpick were considered dead (Appendix A).

The Kaplan–Meier curve represents the mortality of the larvae scored as dead and includes the larvae used for hemolymph extraction at 24 and 48 h (Appendix A). The negative control (PBS only) showed the lowest mortality. By way of contrast, the positive control group showed the highest mortality. All the groups that received ciprofloxacin, except the 1/10th ADI group, had a lower mortality rate than the positive control group, suggesting that these concentrations of ciprofloxacin were having an effect on the *K. pneumoniae*.

### 2.4. Minimum Selective Concentration

The hemolymph of the larvae was cultured onto plates with and without ciprofloxacin, and colonies were automatically counted (Appendix A). The different colony counts for each ciprofloxacin concentration and positive control are provided in Appendix A. The MSC_select_ was defined as the lowest ciprofloxacin concentration, where the proportion of *K. pneumoniae* colonies with resistance was enriched compared to the proportion in the positive control group. This was calculated using the formula (ciprofloxacin/control) to obtain the proportions. The median proportion for each different ADI concentration and the positive control was calculated out of all these proportions (Appendix A).

The proportions of resistance were calculated according to the formula (ciprofloxacin/control). The median was calculated from all the proportions for each different ciprofloxacin ADI concentration and the positive control (Appendix A). Different statistical analyses were performed on the different ciprofloxacin ADI concentrations and the positive control to test their significance (Appendix A).

The median proportion of *K. pneumoniae* colonies with resistance in the positive control group on day 1 was 0.60 (IQR 0.007716; 0.8127) and 0.06 (IQR 0; 0.5225) on day 2 (Appendix A). Furthermore, the median proportion of *K. pneumoniae* colonies with resistance in the 10× ADI group on day 1 was 0.56 (IQR 0.0261; 1.174) and 0.88 (IQR 0.000318; 2.652) on day 2 (Appendix A). No significant difference was seen between these two groups in the median proportions for both days (Appendix A).

For the 1× ADI concentration group, the median proportion of *K. pneumoniae* colonies with resistance was 0.78 (IQR 0.07091; 1) on day 1 and 0.88 (IQR 0.3611; 2.846) on day 2 (Appendix A). No significant increase was found for day 1 compared to the positive control group (Appendix A). However, a significant value of *p* = 0.0233 was found with the Mann–Whitney test, showing an increase in the proportion compared to the control (Appendix A).

The median proportion of *K. pneumoniae* colonies with resistance in the 1/10th ADI group was 0.53 (IQR 0.4338; 2.052) on day 1 and 0.61 (IQR 0.4642; 0.7307) on day 2 (Appendix A). As equal to the ADI concentration, no significant increase was found in the proportion of resistance for day 1 compared to the positive control. However, a significant increase was seen for day 2 in the proportion of resistant colonies (Appendix A). Additionally, a significant value of *p* = 0.0162 was found for day 2 (Appendix A).

Furthermore, the Kruskal–Wallis test found an overall significant *p*-value of 0.0070 for the different ADI concentrations on day 2. This, together with both the significant values of the 1× ADI and 1/10th ADI concentration in the Mann–Whitney test, suggests a difference between these two groups compared to the positive control group. However, the two-stage linear step-up of Benjamini did not find any significance between these two ADI concentrations and the positive control group (Appendix A).

The median proportion of *K. pneumoniae* colonies with resistance in the 1/100th ADI group was 0.15 (IQR 0; 0.6667) on day 1 and 0.072 (IQR 0.004867; 0.6417) on day 2 (Appendix A). No significant increase was seen in the proportion of resistance compared to the positive control for day 1 and 2 (Appendix A). Furthermore, no significant increase was found for the 1/1000th ADI group. Here, the median proportion of *K. pneumoniae* colonies with resistance in the 1/1000th ADI group was 0.46 (IQR 0.09375; 0.8964) on day 1 and 0 (IQR 0; 0.06371) on day 2 (Appendix A).

In summary, a significant increase was seen in the proportion of resistance for the 1× ADI and 1/10th ADI concentration on day 2 compared to the positive control.

### 2.5. E-Tests and Species Identification

The colonies used for E-tests were from plates after the hemolymph extraction on day 1 and day 2. One colony was picked from each of the plates with and without ciprofloxacin from two to three independent larvae for each condition. These colonies were identified at the species level using MALDI-TOF MS (Appendix A).

The MIC distributions (Appendix A) showed that the ciprofloxacin MICs for the *K. pneumoniae* exposed to ADI concentrations were slightly higher than those of the positive controls. This increase was, however, only (just) statistically significant for the 10× ADI group on day 1. On day 1, the medians of the ciprofloxacin plates showed approximately a dilution increase in MIC from 2.5 to 4 mg/L. Conversely, the control plates showed a median increase for the 1× ADI and 1/1000th ADI concentrations. The 1/100th and 1/1000th ADI concentrations had two colonies with an MIC of 6 µg/mL. The positive control contained approximately 50% mutant colonies and 50% wildtype colonies, with one colony that had an increased MIC of 6 µg/mL for the control plates (Appendix A).

On day 1, the biggest variation of MICs was seen for the control plates of the 1/1000th ADI concentration compared to the positive control. However, no variation in MICs was seen on the ciprofloxacin plates between the 10× ADI, 1× ADI, 1/10th ADI, and 1/1000th ADI concentrations. Additionally, the colonies on the control plates of 10× ADI concentration also showed no significant variation. However, a significant value (*p* = 0.0485) was found for the 10× ADI concentration of the ciprofloxacin plates when compared to the positive control (Appendix A).

On day 2, the positive control of the ciprofloxacin plates had a slightly lower median compared to the other ADI concentrations. One exception was the 10× ADI, where no data was obtained. The trend of the positive control was similar to that on day 1. The medians for the control plates were low for the ADI concentrations as well as for the positive control on day 2. Further, higher MICs were seen within the ADI concentrations for the control plates compared to day 1. Specifically, the 1× ADI concentration had an MIC of 6 µg/mL in both the control and ciprofloxacin plates. The 1/10th ADI concentration also showed a maximum MIC of 6 µg/mL on the ciprofloxacin plates. Only on the control plates for the 1/1000th ADI concentration was a maximum MIC of 6 µg/mL seen for both day 1 and day 2 (Appendix A).

## 3. Discussion

To the best of our knowledge, this experiment is the first to attempt to assess an MSC_select_ in vivo. We found that, at 48 h, there was an enrichment of *K. pneumoniae* colonies with ciprofloxacin resistance in the larvae receiving 1× ADI and 1/10th ADI concentrations of ciprofloxacin. These results suggest that the ciprofloxacin MSC_select_ for *K. pneumoniae* in this model is 1/10th of the acceptable daily concentration (ADI) dose of ciprofloxacin, which is equivalent to 0.239 ng/µL.

There are, however, a number of anomalies with our findings. It is surprising that we found evidence of the enrichment of AMR at 1× ADI and 1/10th ADI but not at 10× ADI. This finding could be explained by the smaller number of observations available for the data analysis for the 10× ADI group. This interpretation is compatible with the fact that we found a similar enrichment of AMR in 10× ADI- to 1× ADI and 1/10th ADI groups, but the difference was not statistically significant. Other explanations, such as the microbiome-based explanation outlined in Appendix A, are also possible. Regardless of the underlying mechanism, these findings are similar to those found for the erythromycin MSC_denovo_ for *Streptococcus pneumoniae* in *Galleria mellonella*. This study found that the 1/10th ADI and 1× ADI doses were selected for de novo resistance, whereas the 10× ADI dose did not [7].

A larger anomaly is the fact that we did not observe any enrichment in the day one results. This finding is unexpected as, whilst a number of studies have found that a single dose of ciprofloxacin injected into *G. mellonella* larvae persists for more than 24 h, the highest concentrations would be present in the hours post-injection [6,17,18]. We do not have a clear explanation for this finding. A previous ciprofloxacin MSC_denovo_ study using the same strain of *K. pneumoniae* as the wild type used in this experiment did, however, have compatible results [6]. This study found that single ciprofloxacin doses of as low as 1/10th ADI were selected for de novo ciprofloxacin resistance, but this effect was most pronounced at 48 h post ciprofloxacin administration. Resistant isolates were obtained until day 4.

Further study limitations include the fact that we had a limited number of data points we could use for analysis for a number of the experimental groups. This limited our power to determine statistically significant effects. Future studies would benefit from using larger sample sizes. Furthermore, we only assessed the MSC_select_ for a single bacterial-species–antimicrobial combination. This was only carried out using a single strain of *K. pneumoniae*. These experiments were only performed in *G. mellonella*. Studies are required to assess if the MSC_select_ in this model is similar to those for mammals such as mice and humans [19]. It is plausible that there may be differences between the MSC_select_ obtained in the hemolymph of *G. mellonella* and the gastrointestinal tracts of mammals. We did not consider the interactions between ciprofloxacin and other pharmaceuticals [20]. Finally, we did not evaluate the possible interactions between *K. pneumoniae* and the *G. mellonella* microbiota or assess for the horizontal gene transfer of resistance-conferring DNA to and from the *K. pneumoniae*.

These limitations notwithstanding, our results build on the in vitro MSC_select_ results of others, such as Gullberg et al., that concentrations of ciprofloxacin up to 230-fold below the MIC could select pre-existing resistant strains and enrich these within a population [1]. Our in vivo MSC_select_ is considerably higher than Gullberg et al.’s in vitro MSC. This is not too surprising as other studies have found that the MSC_select_ is higher in complex multi-organism settings than in monocultures such as those used by Gullberg et al. [1]. It should be noted that a further important difference between our experiments and those of Gullberg et al. is that we used *K. pneumoniae*, and they used *E. coli* for their experiments. As noted above, a previous study has found that ciprofloxacin concentrations as low as 1/10th ADI can induce de novo resistance (seven-fold increase in MIC) in *K. pneumoniae* [6]. For *Neisseria gonorrhoeae* and *N. subflava*, the ciprofloxacin MSC_select_ was found to be 1/1000th and 1/100th of the MICs [2,21]. A study conducted by Pereira et al. determined the MSC_select_ in calves by giving them milk containing several antibiotics: ceftiofur, ampicillin, tetracycline, and penicillin [22]. It was found in this study that there was an increase in the proportion of resistant *E. coli* compared to the susceptible strains [22].

These findings are relevant given the growing evidence for peri-MSC concentrations of fluoroquinolones in various environmental and food samples. Concentrations are particularly high in sewage outlets from hospitals and pharmaceutical industries but are frequently still high in soils or aquatic environments [1]. In a lake in India, the ciprofloxacin concentration ranged between 2.5 and 6.5 mg/L [23]. This is concerning because it is higher than the 1/10th ADI dose found in this study. Even more alarming is the fact that the ciprofloxacin concentration ranges between 28,000 and 31,000 µg/L from effluent samples from a wastewater treatment plant containing active pharmaceuticals in India, which is higher than the concentration found to select resistant bacteria [24]. Besides water concentrations, ciprofloxacin is also found within sediments in the Yellow River of China [25]. Here, a concentration of 32.8 µg/kg was found for ciprofloxacin [25]. A survey of pharmaceutical concentrations in the world’s rivers found that the concentration of ciprofloxacin exceeded the proposed safe limit at 64 sites of all the sites sampled [26]. The concentration of ciprofloxacin exceeded the MSC_select_ we found in our study in 3 sites out of the 135 sites.

Numerous foodstuffs consumed by humans contain residual concentrations of ciprofloxacin. The minimum residue limit (MRL) of muscle tissue is set at 100 µg/kg for ciprofloxacin and enrofloxacin by the WHO [27]. However, studies have found that meat frequently contains higher concentrations than the MRL. In Nigeria, raw beef meat contains a mean of 231.08 µg/kg of ciprofloxacin residues, and pork was even higher with a median of 315.30 µg/kg [28]. In China, it was found that the median concentration of ciprofloxacin in edible fish was 331.7 µg/kg [2]. This might suggest that, if a person consumes more than a certain amount of food containing these ciprofloxacin residues in a day, the person might be exposed to concentrations higher than the ADI. Further evidence that low antimicrobial concentrations in food and environmental samples may be contributing to the genesis of AMR in humans comes from ecological studies that have found correlations between antimicrobial concentrations in food/environmental samples and AMR in human-associated bacteria [29].

## 4. Materials and Methods

### 4.1. Bacterial Strain and Live Microbial Inoculum Preparation

The experiments were carried out with two isogenic *Klebsiella pneumoniae* strains (ID M14827). This strain was a clinical isolate from the Institute of Tropical Medicine collection. For further details about this strain, please see [10]. The mutant strain contains an amino acid substitution mutation in the GyrB subunit (Ser464Phe) and has a ciprofloxacin MIC of 4 µg/mL. The wildtype strain has a ciprofloxacin MIC of 0.047 µg/mL.

### 4.2. In Vitro Generation of Mutant Strains and Whole-Genome Sequencing

In vitro generation of mutant strains, WGS, and bioinformatic analysis were carried out at the STI unit, ITM. In brief, the *Klebsiella pneumoniae* strain (ID M14827) was revived out of the frozen stock and subcultured three times overnight onto BD^TM^ Columbia agar with 5% sheep blood. An 0.5 McFarland suspension of the strain was made in PBS from the overnight culture and plated separately onto Mueller–Hinton II agar plates. Next, a ciprofloxacin gradient E-test ranging between (0.002 µg/mL to 32 µg/mL) was placed (AB Biodisk, Stockholm, Sweden). After overnight incubation at 37 °C, a standardized amount of culture that was present closer to the elliptical zone of inhibition was picked and inoculated onto a new Mueller–Hinton II agar plate, followed by placing the E-tests. The procedure was repeated until a colony exhibited a ciprofloxacin MIC of 4 µg/mL. The mutant strain with a ciprofloxacin MIC of 4 µg/mL was stored in skim milk at −80 °C. The E-tests were read using Biomerieux instructions for ciprofloxacin E-test and interpreted according to the EUCAST guidelines with ciprofloxacin resistance at ≥0.125 µg/mL https://mic.eucast.org/search/ (accessed on 10 April 2024).

The mutant strain with a ciprofloxacin MIC of 4 µg/mL was outsourced to Eurofins Genomics (Konstanz, Germany) for isolation of DNA and whole-genome sequencing (WGS) using the Illumina platform. This was followed by bioinformatic analysis and the identification of Ser464Phe substitution at the GyrB subunit. No other relevant mutations were detected within this mutant strain.

### 4.3. Bacterial Suspensions

The mutant (GyrB_Ser464Phe) and wildtype strains were taken out of the frozen stock, subcultured two times onto BD^TM^ Columbia with 5% sheep blood, and incubated overnight at 37 °C with 5% (*v*/*v*) CO_2_. One colony was picked from each subculture on the blood agars and with these two colonies, the 50:50 mix of *K. pneumoniae* bacterial suspension was made. This bacterial suspension was inoculated into the hemocoel of *Galleria mellonella* (10 µL of PBS containing 10^3^ CFU/larva).

### 4.4. Preparation of the 50:50 Mix of the Mutant and Susceptible Inocula for Infection

An equal McFarland stock suspension of the mutant and wildtype was made by picking up colonies from the fresh BD^TM^ Columbia agar plates and suspending them into PBS. Next, an equal volume was pipetted from these solutions to make the 50:50 mix. This mix was vortexed thoroughly to make a homogenous suspension from both strains. A spectrophotometer (Genesys 20, Thermo Spectronic, San Francisco, CA, USA) was used to validate the concentrations of the two stock solutions before mixing and after the 50:50 mix at OD_600nm_. Thus, the 50:50 mix was made with both stock suspensions having approximately the same OD_600_.

The 50:50 ratio of the *K. pneumoniae* mix containing the susceptible and mutant strains in equal proportions was validated in vitro. Separate stock suspensions were made from the mutant and wildtype strains. Then, the 50:50 mix was made by pipetting an equal volume of the stock suspensions. The bacterial concentrations were validated to be an equal amount within the stock solutions and 50:50 mix using a spectrophotometer. Further, serial dilution passages with concentrations of 10^−5^ and 10^−6^ were prepared for the wildtype, mutant, and 50:50 mix and cultured onto *Klebsiella* ChromoSelect plates with and without ciprofloxacin (control). The concentrations of 10^−5^ and 10^−6^ were selected based on their observable colony density, in that these concentrations exhibited countable forming units. The serial dilution passage concentrations were validated with a spectrophotometer at OD_600 nm_.

### 4.5. Injection of the Inoculum in the Galleria mellonella Larvae

The last right pro-leg of *G. mellonella* (Terramania, Arnhem, The Netherlands) was injected with the 50:50 bacterial mix suspension. The injection was carried out with only non-discolored, healthy larvae weighing between 250 to 450 mg. For the injection, insulin syringe needles of 0.3 mL U-100 (BD Micro-Fine, Brisbane, Australia) were used to inject 30 µL of bacterial mix suspension (2.73 × 10^5^ CFU/mL) into the last right pro-leg. After 10–20 min, 10 µL of the ciprofloxacin ADI equivalent dose was injected into the left pro-leg. One insulin syringe needle was used for ten larvae. After injection, the larvae were placed in the incubator during the experiments at 37 °C in a 5% (*v*/*v*) CO_2_ atmosphere. Ten larvae were picked per Petri dish, and 30 larvae were used for each ciprofloxacin ADI dose (10× ADI, 1× ADI, 1/10th, 1/100th, and 1/1000th ADI dose). Further, ten larvae were used for negative control with PBS, and five larvae were injected with only the antibiotic condition as the negative control. For the positive control, 30 larvae were injected with only the 50:50 mix and 10 µL PBS.

### 4.6. Injection of Ciprofloxacin

Single doses of ciprofloxacin were injected into the larvae based on the acceptable daily intake of 6.2 µg/kg of enrofloxacin. This dosage was adapted for the larvae based on the average weight of 380 mg. An ADI dose concentration of 2.36 ng of ciprofloxacin was used per larvae. To obtain this concentration of 2.36 ng in vivo in the larvae, different ciprofloxacin doses (required concentration) were calculated based on the liquid volume and weight of the larvae (Appendix A). These required concentrations of ciprofloxacin that needed to be injected were calculated based on the formulae from a study by Andrea et al. [30]. This was carried out by randomly weighing 20 larvae used per batch, and the average weight was calculated based on these weights. First, the correlation equation of the study from Andrea et al. was used to compare the larva’s liquid volume and total weight.

The liquid volume (y) was calculated by filling out the average weight (x) of the 20 individual larvae in the equation below (Appendix A). Next, the concentration of the injected ciprofloxacin was calculated using a different formula found by Andrea et al.
(1)$$C_{compound}=\frac{C_{in\;vivo}\times V_{larva}}{10\;µL}$$

The V_larva_ component was calculated by the sum V_liquid_ + V_infection_ + V_compound_. The V_liquid_ was calculated using the equation y = 0.6102x + 9.4889. The concentration in vivo for the larvae was 2.36 ng of ciprofloxacin calculated from the ADI of 6.2 µg/kg. The required antibiotic concentration (C_compound_) was calculated and filled out in the dilution scheme to make the correct antibiotic dilution containing the correct concentration based on their weight (Appendix A). This dilution scheme was calculated using the formula C_1_ × V_1_ = C_2_ × V_2_, where C_2_ represents the new calculated concentration (required concentration), according to the formulae mentioned above, and C_1_ is the stock solution of ciprofloxacin. V_1_ contains the volume needed to take out of the stock solution of ciprofloxacin, and V_2_ is the total volume. Milli Q water was used to dilute the concentrations. Different ciprofloxacin concentrations were calculated for each batch of larvae to inoculate the different ADI concentrations (Appendix A).

### 4.7. Retrieval of K. pneumoniae from G. mellonella

The mortality rate of the larvae was determined at 24 and 48 h intervals after the injection by a scoring system, where they were scored as being dead or alive. Larvae that did not respond to the touch of a toothpick and showed strong melanization were considered dead. The number of larvae used for the extraction was adjusted according to larval mortality (60–80%) such that the range of larvae sacrificed was between 4 and 10 at 24 and 48 h time points. An incision was made in the last segment of the larvae, and the hemolymph was squeezed to collect in an Eppendorf tube containing 75 µL of PBS [6]. This was vortexed thoroughly, and 50 µL was plated out onto the *Klebsiella* ChromoSelective agar plate (KA, Merck [Darmstadt, Germany]) without ciprofloxacin (control) and on the ChromoSelective agar plate containing ciprofloxacin (1 µg/mL) to deduce the ratio of the wildtype vs. mutant colonies. After the hemolymph extraction, all the larvae were kept in the freezer at −80 °C, immobilizing and killing them. Next, the larvae were autoclaved at 121 °C for 15 min and disposed of. The plates containing the hemolymph were incubated at 37 °C with a 5% (*v*/*v*) CO_2_ atmosphere for 24 h, and the colored purple–magenta colonies were automatically counted using SCAN 300^®^ (Interscience, Cantal, France), version 8.7.3.0. Several colonies were picked out at random for E-testing and identification via MALDI-TOF-MS (Bruker Daltonics, Bremen, Germany).

### 4.8. MSC_select_ Determination

The MSC_select_ in vivo of ciprofloxacin is determined based on the increase in proportion of the mutant (resistant) vs. the susceptible colonies. For each experimental condition, a single larva was used to extract the hemolymph in 75 µL of PBS, followed by plating onto the *Klebsiella* ChromoSelect plate with and without ciprofloxacin. This was carried out for several different larvae for each condition (10× ADI, 1× ADI, 1/10th, 1/100th, and 1/1000th ADI dose, and positive control). The colonies were counted on each plate using an automatic colony counter SCAN 300^®^ (Interscience, Cantal, France), version 8.7.3.0. The proportion of the mutant colonies vs. the susceptible colonies was then calculated.

For each ADI concentration, one colony was picked from the plate with ciprofloxacin and one colony from the plates without ciprofloxacin to determine their MICs. The colonies were suspended in PBS until 0.5–0.6 McF was reached. Afterwards, the suspension was plated onto Mueller–Hinton II agar plates using a sterile swab, and E-test was placed on the agar plates. After a 21 h incubation, the E-tests were read according to the EUCAST guidelines.

### 4.9. MALDI-TOF Mass Spectrometry

Matrix-assisted laser desorption/ionization–time-of-flight mass spectrometry was used to identify the purple–magenta colonies picked for determination of the MIC. The species conformation happened on a MALDI Biotyper Sirius IVD system using the MBT Compass IVD software V1.4 and library (Bruker Daltonics, Bremen, Germany). One single colony was picked up with a toothpick and put onto a reusable polished metal MALDI-TOF plate until dried. The internal control, BTS, was spotted in the first spot. Onto the colony spots, 1 µL of formic acid was added. After, 1 µL of α-cyano-4-hydroxycinnamic acid (CHCA) matrix solution was spotted on the spots containing BTS and colonies. Next, the plate was loaded into the mass spectrometer, and the spots were read. This was carried out in a linear mode in a mass range of 2–20 kDa. The spectra were compared to a library containing known spectra. For the identification, reliability cut-off values of 1.7 and 2 were used to validate the results as reliable or unreliable according to their genus and species (Appendix A).

### 4.10. Data Processing

Plates with a large number of colonies that were uncountable were adjusted for in the dataset. The highest number of colonies that could be counted by the colony counter on a plate was 3147. All plates with more than 3147 colonies were classified as having 3147 colonies. Given this imprecision, when both plates with and without ciprofloxacin had 3147 colonies, these were dropped from further analyses. In addition, data were dropped if the ciprofloxacin plate (the denominator) had 0 colonies (x/0). The proportions of each set of plates, control and ciprofloxacin, were calculated according to the formula (ciprofloxacin/control).

### 4.11. Data Analysis

Statistical analyses and data visualization, such as graphs and boxplots, were performed using GraphPad Prism^®^ version 9.5.1. The Kruskal–Wallis test was used to assess if there was a difference in the proportion of ciprofloxacin-resistant colonies in any of the groups that received ciprofloxacin compared to the group that received PBS. A two-stage linear step-up procedure of Benjamini, Krieger, and Yekutieli was used to adjust for multiple comparisons. The Kaplan–Meier statistical method was used for survival analysis. *p*-value < 0.05 was considered statistically significant. Visual schemes of materials and methods were created with BioRender.com.

## 5. Conclusions

Previous studies have confirmed that antibiotic residues could select for antimicrobial resistance in bacteria even at concentrations lower than the MIC. The results in this study show the enrichment in resistant colonies compared to the susceptible colonies of *K. pneumoniae* at concentrations of 1× ADI and 1/10th ADI of ciprofloxacin on day 2. This suggests that the MSC_select_ is 1/10th the ADI dose of ciprofloxacin. This means that ciprofloxacin concentrations as low as 1/10th the ADI dose or 0.239 ng/µL could select for the resistant *K. pneumoniae* strain. Future research should evaluate if this effect is evident using mammalian models. In addition, the synergistic effects of antimicrobials with other pharmaceuticals (such as selective serotonin reuptake inhibitors) or heavy metals should be evaluated. In conclusion, our findings contribute to those from other studies that suggest the need to incorporate the MSC_select_ and MSC_denovo_ in establishing safe antimicrobial concentrations in food (ADIs and MRLs).

## Figures and Tables

**Table 1 antibiotics-13-01097-t001:** Statistical analysis was performed on the wildtype and mutant strain compared to the 50:50 *K. pneumoniae* mix for the concentrations 10^−5^ and 10^−6^ on the control and ciprofloxacin plates.

**Concentration 10^−5^**		
Mann–Whitney test—ciprofloxacin plates	Significant	*p*-value
Mix vs. Mutant	Yes	0.0079
Mix vs. Wildtype	Yes	0.0079
Mann–Whitney test—control plates	Significant	*p*-value
Mix vs. Mutant	No	0.2143
Mix vs. Wildtype	No	0.5476
**Concentration 10^−6^**		
Mann–Whitney test—ciprofloxacin plates	Significant	*p*-value
Mix vs. Mutant	No	0.1746
Mix vs. Wildtype	Yes	0.0079
Mann–Whitney test—control plates	Significant	*p*-value
Mix vs. Mutant	No	0.2063
Mix vs. Wildtype	No	0.0952

Mann–Whitney test was carried out to determine significant differences between the two strains compared to the 50:50 mix.

**Table 2 antibiotics-13-01097-t002:** Statistical analyses of the different ciprofloxacin ADI concentrations and the positive control.

**24 h Extraction (Day 1)**				
Mann–Whitney test		Significant	*p*-value	
Positive control vs. 10×		No	0.1118	
Positive control vs. 1×		No	0.1118	
Positive control vs. 1/10th		Yes	0.0040	
Positive control vs. 1/100th		No	0.2560	
Positive control vs. 1/1000th		Yes	0.0366	
Kruskal–Wallis test				
*p*-value	0.0005 < 0.05			
Two-stage linear step-up of Benjamini		Significant	q-value	Individual *p*-value
Positive control vs. 10×		No	0.2091	0.1593
Positive control vs. 1×		No	0.2091	0.1395
Positive control vs. 1/10th		No	0.0888	0.0169
Positive control vs. 1/100th		No	0.3887	0.3702
Positive control vs. 1/1000th		No	0.0901	0.0343
**48 h Extraction (Day 2)**				
Mann–Whitney test		Significant	*p*-value	
Positive control vs. 10×		No	0.2930	
Positive control vs. 1×		No	0.9845	
Positive control vs. 1/10th		Yes	0.0378	
Positive control vs. 1/100th		Yes	0.0105	
Positive control vs. 1/1000th		No	0.0825	
Kruskal–Wallis test				
*p*-value	0.0893 > 0.05			
Two-stage linear step-up of Benjamini		Significant	q-value	Individual *p*-value
Positive control vs. 10×		No	0.2322	0.1769
Positive control vs. 1×		No	>0.9999	0.9825
Positive control vs. 1/10th		No	0.0864	0.0329
Positive control vs. 1/100th		No	0.0989	0.0565
Positive control vs. 1/1000th		No	0.0864	0.0254

The Mann–Whitney test was performed with the different ciprofloxacin ADI concentrations against the positive control. Two significant values were found for 1/10th ADI (*p* = 0.0040) and 1/1000th ADI (*p* = 0.0366) on day 1. The Kruskal–Wallis test was used to test the overall difference between all the different groups. A significant *p*-value of *p* = 0.0005 < 0.05 was found for day 1. The two-stage linear step-up of the Benjamini test is used to adjust for multiple comparisons. Two significant values were found for day 2, the 1/10th ADI concentration (*p* = 0.0378) and the 1/100th ADI concentration (*p* = 0.0105).

## Data Availability

All data are provided within the article and Appendix A.

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
