# Peer review of "Ciprofloxacin Concentrations in Food Could Select for Quinolone Resistance in Klebsiella pneumoniae: An In Vivo Study in Galleria mellonella"

_antibiotics, 2024, doi:10.3390/antibiotics13111097_

Round 1
Reviewer 1 Report
Comments and Suggestions for Authors
Ciprofloxacin concentrations in food could select for quinolone resistance in Enterobacteriaceae: an in vivo study in Galleria mellonella
The manuscript is reviewed for consideration in Journal. The topic is very interesting and methodology adopted is appropriate. However, references cited in discussion needs to be double check with statements, and following point/suggestions should be included.
Introduction
It is most desirable that the last paragraph should contain the uniqueness and innovation in your study. This must be important to highlight the importance of study.
Methodology
Bacterial strains are not completely briefed with comprehensive methodology.
Conclusion
The conclusion part is missing, should include a comprehensive conclusion
Reviewer 2 Report
Comments and Suggestions for Authors
Overall, this study has demonstrated proven quality and applicable knowledge that can significantly impact and improve the WHO and other regulatory authorities' consideration of the minimum selection concentration (MSC) values when calculating the acceptable daily intake (ADI) and the maximun residue limits (MRL) values. The methodological approach is also satisfactory, particularly the experimental design. It should be noted that the small sample size is a limitation of this study.
Comparisons of results obtained from different experimental conditions, which a priori appear to be non-significant, would probably have been different if the data points analyzed were more representative.
However, the conclusions are reliable; therefore, the article can be accepted in its current form subject to minor modifications.
Minors corrections:
1- Line 208, please remove the space before "A two-stage...",
2- Lines 215-224: This part is more related to the validation of the method used in the experimental design. Therefore, for a better structuring of the paper, it will be necessary to move it to the methodology section,
3- Line 258, please remove the space before "2 were counted..".
Reviewer 3 Report
Comments and Suggestions for Authors The study addresses an interesting topic, namely the development of antibiotic resistance after intake of minimal doses of an antibiotic (ciprofloxacin) found as residues.The study is important and original. It answers questions asked in other articles on food and residual antibiotic substances and confirms theories about the influence of the antibiotics in food.
The methodology is described in detail and adequately, confirming the team's expertise. It allows other authors to replicate the study.
The results are described in detail and well illustrated.
The conclusions are appropriate to the aim and the obtained results.
The study occupies an important place in the scientific field as it is the first to study MSCselect in vivo. I don't find any flaws in the methodology.
I would suggest to the authors:
1. It would be better if some abbreviations are given with full name the first time they are mentioned (MIC, CAC, WGS, PBS, STI and etc.).
2. The names of the bacteria should be written in full the first time they are mentioned (E. coli, K. pneumoniae).
3. Line 66: "6,2 ug/kg or ug/kg/bw"
4. As the study includes only Klebsiella pneumoniae, the title can be changed to include only this microorganism instead of the family name.
